



# A new lightning scheme in Canada's Atmospheric Model, CanAM5.1: Implementation, evaluation, and projections of lightning and fire in future climates

Cynthia Whaley[1], Montana Etten-Bohm[2,3], Courtney Schumacher[3], Ayodeji Akingunola[1], Vivek Arora[1], Jason Cole[1], Mike Lazare[1], David Plummer[1], Knut von Salzen[1], and Barbara Winter[1]

[1]Canadian Centre for Climate Modelling and Analysis, Environment and Climate Change Canada, Victoria, BC, Canada.
[2]Department of Atmospheric Sciences, University of North Dakota, USA.
[3]Department of Atmospheric Sciences, Texas A&M University, College Station, TX, USA.

**Correspondence:** Cynthia Whaley (cynthia.whaley@ec.gc.ca)

**Abstract.** Lightning is an important atmospheric process for generating reactive nitrogen, resulting in production of tropospheric ozone, as well as igniting wildland fires, which result in potentially large emissions of many pollutants and short-lived climate forcers. Lightning is also expected to change in frequency and location with the changing climate. As such, lightning is an important component of Earth system models. Until now, the Canadian Earth System Model (CanESM) did not contain

an interactive lightning parameterization. The fire parameterization in CanESM5.1 was designed to use prescribed monthly climatological lightning. In this study, we have added a logistical regression lightning model that predicts lightning occurrence interactively based on three environmental variables and their interactions into CanESM5.1's atmospheric model, CanAM5.1, creating the capacity to interactively model lightning, allowing for future projections under different climate scenarios. The modelled lightning and resulting burned area were evaluated against satellite measurements over the historical period and model

biases were found to be acceptable. Modelled lightning was within a factor of two of the measurements and had exceptionally accurate land/ocean ratios.

The modified version of CanESM5.1 was used to simulate two future climate scenarios (SSP2-4.5 and SSP5-8.5) to assess how lightning and burned area change in the future. Under the higher emission scenario (SSP5-8.5), CanESM5.1 predicts an increase in northern mid-latitude lightning flashrate of 5%, but a decrease in tropical lightning of -10%, resulting in almost

no change to the global mean lightning amount by the end-of-the century (2081-2100 vs 2015-2035 average). By century's end, the change in global total burned area with prescribed climatological lightning was about two times greater than that with interactive lightning (43% vs 19% increase, respectively). Conversely, in the northern mid-latitudes the use of interactive lightning resulted in three times more area burned as that with unchanging lightning (36% vs 13% increase, respectively). These results show that the future changes to burned area are greatly dependent on a model's lightning scheme, both spatially

and overall.



# 1 Introduction

In addition to being a hazard to human health (Jensen et al., 2022) and infrastructure (Mills et al., 2010), lightning is an indirect source of short-lived climate forcers (SLCFs) given that it produces nitrogen oxides ($NO_x$) – a tropospheric ozone ($O_3$) precursor – in the atmosphere, and it is responsible for igniting wildland fires, which in turn emit numerous greenhouse

gases and SLCFs, such as methane ($CH_4$), black carbon, and $O_3$ precursors (e.g., $CH_4$, volatile organic compounds [VOCs] and $NO_x$). Lightning is also expected to increase with climate change in several regions, though studies differ greatly depending on which lightning parameterization and model is used (Williams, 2005; Zeng et al., 2008; Hui and Hong, 2013; Price, 2013; Krause et al., 2014; Banerjee et al., 2014; Clark et al., 2017; Finney et al., 2018; Chen et al., 2021).

Lightning is also difficult to model accurately. The processes responsible for cloud electrification are not simulated in current
climate models, so parameterizations of lightning have relied on associations between lightning and large-scale or uncertain variables, like cloud height. However, many lightning schemes do not reproduce the observed ocean-land gradient and/or need separate parameters over land vs. over ocean (e.g., Murray et al., 2012; Romps et al., 2018). To reliably project changes to lightning and wildfires in the future, as well as to better understand their interactions, it is imperative to realistically simulate lightning in Earth system models.

Several lightning parameterizations are available for use in atmospheric models, each with its own benefits and drawbacks. For example, the Price and Rind (1992, 1993) lightning scheme, based on cloud-top height, is popular in climate models due to its computational efficiency, but exhibits poor skill (Tost et al., 2007; Murray et al., 2012). The Finney et al. (2014) and Allen and Pickering (2002) schemes are based on upward ice flux at 440 hPa, and show better results than cloud-top height models over the oceans. The Lopez (2016) lightning scheme is based on charging rate and convective available potential energy
(CAPE), but it requires graupel, snow, and cloud condensate in updrafts, and these are often not explicitly simulated in most atmospheric models. Similarly, McCaul et al. (2009) have a scheme based on upward flux of graupel and the integral of solid precipitate. However, because it too, requires the explicit simulation of microphysics for cloud water, snow, and graupel, it is only appropriate for very high horizontal resolution models. Finally, the lightning scheme based on the product of CAPE and precipitation (Romps et al., 2014) has gained recent attention, but this, too, has been shown to perform poorly over ocean
(Romps et al., 2018).

Projected lightning changes with climate differ greatly depending on the lightning parameterization and the underlying model. For example, in the tropics, lightning has been projected to increase based on cloud-top schemes, but decrease based on ice-flux schemes (Finney et al., 2018). Generally, lightning is projected to increase in the northern mid-latitudes (e.g. Janssen et al. (2023)) and even in the Arctic where it was previously non-existent (Chen et al., 2021), but this continues to be a highly
uncertain projection.

In this study, we evaluated a logistic regression lightning model from Etten-Bohm et al. (2021) in version 5.1 of Canada's Earth System Model, CanESM5.1 (Sigmond et al., 2023). The Etten-Bohm et al. (2021) lightning scheme has the benefit of a single formulation that works well over both land and ocean. It depends on well-known environmental variables that atmospheric models compute routinely and it doesn't require tuning to a global mean value. In Section 2 we describe this





**Table 1.** Fitted coefficients in the lightning model from Etten-Bohm et al. (2021) used in Equation 1.

| Coefficient | Corresponding Variable | Value from fit |
|:---:|:---:|:---:|
| $B_0$ | | -6.3509 |
| $B_1$ | CAPE | 0.779 |
| $B_2$ | LCL | -1.303 |
| $B_3$ | r | 1.230 |
| $B_4$ | CAPE*LCL | -0.360 |
| $B_5$ | CAPE*r | -0.050 |
| $B_6$ | LCL*r | -0.167 |

lightning scheme, its implementation in CanESM5.1, and its subsequent evaluation. Section 3 shows the modelled lightning and burned area results and their comparisons to observation-based datasets. In Section 4, we report results from future CanESM5.1 simulations with the new interactive lightning scheme to the end of the century to examine how lightning and area burned change in future climate scenarios. Finally, conclusions are presented in Section 5.

## 2 Methods

### 2.1 Lightning model

The lightning parameterization we have selected for use in CanESM5.1 was derived in Etten-Bohm et al. (2021), where the relationship between lightning and several large-scale environmental variables were assessed. We use Etten-Bohm et al. (2021)'s "model b", which provided the best lightning results based on three environmental variables, i.e., undilute CAPE, lifting condensation level (LCL), and column saturation fraction ($r$), and their interactions to determine the probability ($p$) of a lightning occurrence at grid point $s$ (Equation 1):

$$\text{logit}(p(s)) = \log \frac{p(s)}{1 - p(s)}$$
$$= B_0 + B_1 \text{CAPE}(s) + B_2 \text{LCL}(s) + B_3 \text{r}(s) + B_4 \text{CAPE}(s) \times \text{LCL}(s) + B_5 \text{CAPE}(s) \times \text{r}(s) + B_6 \text{LCL}(s) \times \text{r}(s) \quad (1)$$

where the $B_i$ coefficients are given in Table 1 and LCL is in pressure coordinates (in mb). The coefficients were determined through a logistic regression, which was trained on one year (2003) of 0.5° gridded lightning data from the Tropical Rainfall Measuring Mission (TRMM) Lightning Imaging Sensor (LIS) and environmental variables from the Modern-Era Retrospective analysis for Research and Applications Version 2 (MERRA-2), and tested with data from 2004.

As discussed in Section 1, CAPE has been used in some lightning parameterizations because of its strong link to a storm's potential updraft intensity. The column saturation fraction, $r$, is a measure of how humid a column is relative to its saturation specific humidity, and is analogous to column water vapour. $r$ has been shown to be highly related to convective precipitation,



especially over tropical oceans (Bretherton et al., 2004) but also over warm land regions (Ahmed and Schumacher, 2017). LCL and related proxies have been shown to help distinguish between land and ocean lightning occurrence because moister areas, like over the ocean, tend to have lower LCLs, and therefore lower cloud bases, which has been linked to less lightning (Etten-Bohm et al., 2021; Stolz et al., 2015; Williams and Stanfill, 2002).

## 2.2 CanESM5.1

We implement the above lightning scheme as a new subroutine in the physics module of CanAM5.1 (Cole et al., 2023), the atmospheric model component of CanESM5.1 (Swart et al., 2019). The operational horizontal resolution of CanESM5.1 is T63 ($\sim 2.8°$) resolution in the atmosphere, and $\sim 1°$ in the ocean which is based on Nucleus for European Modelling of the Ocean (NEMO, Madec and the NEMO team (2012)) model. The land component of CanESM5.1 is based on "Canadian Land Surface Scheme"-"Canadian Terrestrial Ecosystem Model" (CLASS-CTEM), which simulates area burned and fire $CO_2$ emissions (Arora and Melton, 2018; Arora and Boer, 2005). The emissions of several other species are based on specified emissions factors. In CanESM5.1, atmospheric pollutant concentrations from fires are specified based on the CMIP6 protocol (Verseghy, 1991; Verseghy et al., 1993; Verseghy, 2000; Arora, 2003; Arora and Boer, 2003, 2005; Swart et al., 2019), that is, from input emissions. The linkage between CLASS-CTEM fire emissions of various species and the atmospheric aerosols module is not made in this study, but will be a subject of future work.

Figure S0 in the supplement explains how CanAM5.1 computes CAPE within its convection subroutines. In this formulation, CAPE is defined as negative when the air parcel moves downward. We adjust this for the lightning calculation such that CAPE input for lightning is only positive for upward moving parcels, and zero otherwise. LCL is also calculated as a vertical index in the same subroutine as CAPE and is passed to the new lightning subroutine where the pressure at that index is used for the lightning calculation. $r$ is a new calculation in CanESM5.1, based on the model's specific humidity and saturation mixing ratio (Ahmed and Schumacher, 2017).

New model outputs include lightning occurrence (given as a percent probability), the total lightning flash rate (given in flashes/km$^2$/year), and the cloud-to-ground and cloud-to-cloud flash rates. Lightning occurrence is calculated from Equation 1 and the total flash rate is calculated using the product of a scale factor and lightning occurrence. Etten-Bohm et al. (2021) showed that the mean lightning occurrence and mean flash rates observed by the TRMM LIS have very similar geographical patterns, so the scale factor was found by determining the multiplicative factor that results in a global average flash rate that is similar to that observed by TRMM LIS/Optical Transient Detector (hereafter "LIS/OTD"). The cloud-to-ground fraction was set to a linearly increasing value based on latitude, with 10% fraction at the equator, increasing to 50% at the poles, based on observations of the freezing height in the clouds and resulting cloud-to-ground fraction (Uman, 1986).

### 2.2.1 Fires in CanESM5.1

In the simulation with interactive lightning, the cloud-to-ground lightning flash rate is used in place of the specified climatological lightning for natural wild fire ignition in CLASS/CTEM. The specified climatological lightning is based on LIS/OTD total lightning flash rate, pre-converted to cloud-to-ground fraction for the input file. CLASS/CTEM's fire module also has a





human ignition and suppression component (Arora and Melton, 2018), which is based on population density. In this study, we
used an unchanging present-day human population density corresponding to the average of 2010-2019.

The fire module in CLASS/CTEM is designed to capture large-scale global fire behaviour and, in addition to lightning, is dependent on simulated vegetation biomass and soil moisture. The fire module calculates probability of fire based on availability of biomass as a fuel source, combustibility of fuel based on its moisture content, and the presence of an ignition source (be it human or lightning). Since CTEM, the biogeochemistry component, operates at a daily time step, area burned is calculated daily. The area burned in one day is based on probability of fire, wind speed, and the fire duration which is expressed
in terms of the fire extinguishing probability. Fire extinguishing probability in turn is dependent on human population density. CLASS/CTEM fire emissions and burned area have been evaluated when the model is driven offline (driven by bias-corrected climate input) (Li et al., 2019), and by reanalysis data (Arora and Melton, 2018). However, CLASS-CTEM's area burned estimates have not been evaluated within the CanESM5.1 framework before this study.

Note that two preindustrial spin-up simulations (one with prescribed lightning and one with interactive lightning) of CanESM5.1 were conducted for 150 years each in order for the global vegetation to equilibrate after having fire turned on the first time. Then the transient historical simulations of this study were performed, with 10 ensemble members, starting from 1850 for additional historical spin-up time, where we keep the results from 1995 onward for evaluation and analysis.

### 2.2.2 Future simulations

We simulate the future time period (2015 to 2100) with 10 ensemble members for two future climate change scenarios: the severe shared socioeconomic pathways (SSP5-8.5) and the moderate (SSP2-4.5) (Riahi et al., 2017). We average model results over the last twenty years (2081-2100) of the future scenarios and compare them to the average of the first twenty years (2015-2035, representing the present).

### 2.2.3 Evaluation

In Section 3, we evaluate lightning occurrence, flash rate, and burned area against the following observation-based datasets, for different groupings of years:

- the International Space Station (ISS) Lightning Imaging Sensor (LIS), hereafter "ISS LIS", lightning occurrence dataset, which covers from 54°S to 54°N and started on 1 March 2017 (Blakeslee et al., 2020). We evaluate the years 2017-2019,

- the gridded climatology of total lightning flash rate from the spaceborne OTD and TRMM LIS (same as LIS/OTD
mentioned above) (Cecil et al., 2014) (we evaluate 1995-2014), and

- the MODIS fire_cci v5.1 area burned product (evaluate 2001-2014), (Lizundia-Loiola et al., 2020)

Note that while the LIS/OTD flash rate is a global product, its OTD data were collected from May 1995 to March 2000, and its TRMM LIS data (equatorward of about 38°) are from 1998 to 2014. Thus, the LIS/OTD climatology is most robust in the tropics and subtropics, while the high-latitude data is entirely from OTD (Cecil et al., 2014). CanAM5.1's CAPE, LCL, and





r were compared to those from the MERRA-2 renanalysis (Section 3.2 and in the Supplement), informing the results of the lightning evaluation.

## 3    Evaluation Results

Here we show the total (which includes cloud-to-ground and cloud-to-cloud) lightning results from the CanESM5.1 simulation that contains the Etten-Bohm et al. (2021) lightning scheme, and compare those results to both the ISS LIS lightning occurrence

dataset (Section 3.1) for 2017-2019 (Figures 1 and 2), and to the LIS/OTD lightning flash rate climatology for 1995-2014 (Section 3.3, Figures 4 and 5). We also evaluate the burned area (Section 3.4) against MODIS-derived data for 2001-2014, as modelled burned area is impacted by the cloud-to-ground component of the new lightning.

### 3.1    Lightning occurrence

The annual average lightning occurrence, given as a percent that lightning occurs in each model column, is evaluated using the

54°S to 54°N observations from the ISS LIS instrument. The ISS LIS observations were first interpolated onto the model grid. These geographical distributions are compared in Figure 1, and the zonal means and the seasonal cycle (regionally averaged monthly means) are compared in Figure 2. The global mean absolute difference between model and observations is -0.5%.

The spatial distribution of lightning occurrence shows that our model configuration results in near-accurate land-ocean differences in lightning, with very little lightning over the ocean. The land/ocean ratio of our modelled lightning occurrence is

3.0 when the whole globe is considered, and is 5.2 when only 54°S to 54°N are considered. The latter can be more directly compared to the land/ocean ratio from ISS LIS, which is 5.0. That ratio in other models is often less than 1 (Charn and Parishani, 2021). This is already a large advantage over other lightning schemes mentioned in the introduction, and consistent with the Etten-Bohm et al. (2024) results, where this lightning scheme was implemented in the CAM5 model.

However, over the western coasts of North and South America, the modelled lightning is significantly higher than that

observed. One feature noted during development was that this lightning scheme has resulted in too much lightning over the mountains. In Etten-Bohm et al. (2021) and Etten-Bohm et al. (2024), the mountainous regions with elevation greater than 1500m were removed from their analysis and figures. In an effort to improve this aspect, we removed the primary LCL term in Equation 1 for model grid cells with topography elevation greater than 1500 m, and this resulted in less lightning over the North American Cordillera and over the Himalayas. The results shown in this paper include this adjustment, and still have the

overestimation over mountains that are > 1500 m.

Conversely, the modelled lightning is biased low over the eastern half of North and South America and in India. In Section 3.2, we will see that these low biases correspond spatially to negative biases in CAPE for the former, and LCL and $r$ for the latter. In Africa, there is a negative bias in the northern half and a positive bias in the southern half. Those regional biases are consistent with all of the CAPE, LCL, and $r$ biases (Section 3.2). Finally, Australia has a positive bias, where LCL and $r$ are

biased positive as well. In order to improve CanESM lightning, improvements to the underlying parameters are needed. In this



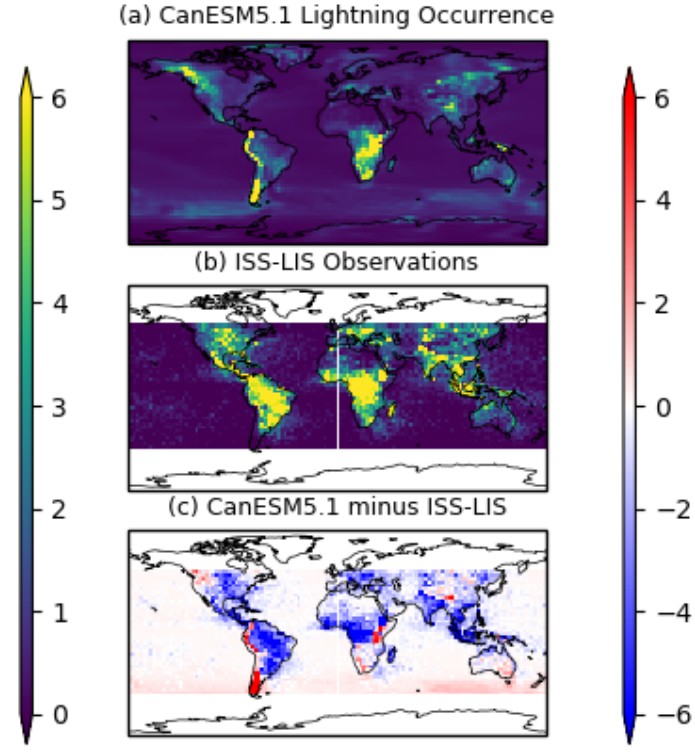

**Figure 1.** Comparison of the 2017-2019 mean modelled lightning occurrence from CanESM5.1 to that measured by the ISS LIS instrument, and their absolute difference (all in %).

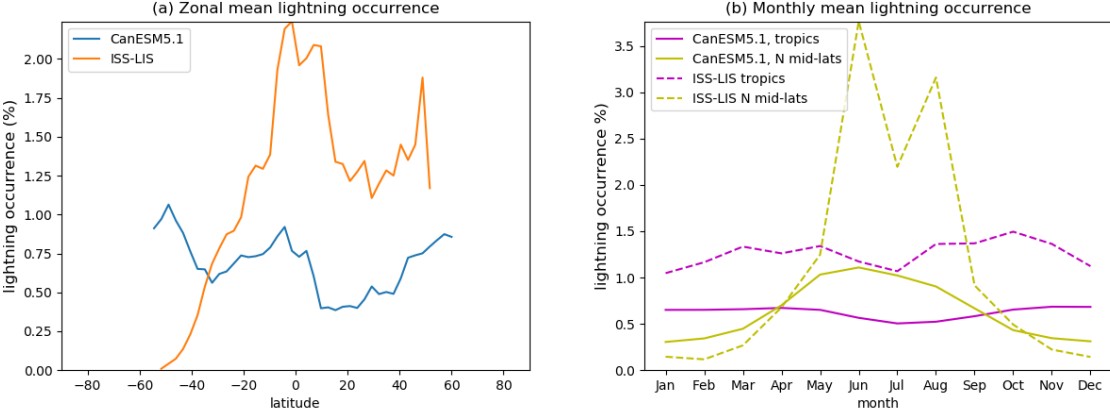

**Figure 2.** (a) Comparison of zonal mean lightning occurrence from CanESM5.1 and that measured from ISS LIS for 2017-2019. (b) Comparison of the seasonal cycle in the lightning occurrence from CanESM5.1 (solid lines) and that measured from ISS LIS (dashed lines) for 2017-2019 in the tropics (30°S-30°N) and northern hemisphere mid-latitudes (30°N-54°N).





vein, work is underway to use a new TKE deep convection scheme for future versions of CanAM(5.2+), and lightning can be reassessed when that is in place.

The zonal mean in Figure 2a highlights that the modelled lightning is also biased high over southern latitudes and low over northern latitudes. But from about 30°S to 50°N, the zonal pattern is modelled correctly. Note that 54°S and N are the maximum bounds of the ISS LIS observations.

Finally, the regional seasonal cycles of lightning occurrence are shown in Figure 2b for the average within defined latitude bands. We define the tropics as the mean between 30°S to 30°N, and the mid-latitudes as 30°N to 54°N. Aside from the systematic offset, the modelled lightning seasonality for the tropics is similar with minimum in the summertime and increases in the spring and fall. Both model and observations have a summertime peak in the northern mid-latitude lightning occurrence as well, though the model peak is wider than observed. The southern mid-latitudes (not shown) have a seasonal peak in Dec and Jan and minimum in July for both the model and measurements. Since ISS LIS can't observe the Arctic region, and the southern mid-latitudes have little land, we don't include those in Figure 2b. We will see in the next section that the high-Arctic (>75°) model results should not be considered.

## 3.2 Evaluation of input parameters

CanESM5.1's CAPE, LCL, and $r$ (the lightning input parameters) are evaluated against those computed from the MERRA-2 reanalysis, with figures shown in the supplement and a summary of their biases in Figure 3. CanESM5.1 underestimates CAPE (Figures 3a, S1 and S2), except in the southern ocean and north Atlantic ocean regions, where it is overestimated slightly. It is likely that the positive CAPE bias in the southern ocean contributes to the increased lightning in that region. The negatively biased CAPE in the tropics is also likely the reason why CanESM's lightning occurrence is biased low there too. CanAM5.1's low CAPE bias has been documented previously in Mitovski et al. (2019), where it was found to be about three times too low in the tropics, consistent with the findings here. However, because the environmental variables are standardized around their mean value before being input into the logistic regression (Equation 1), overall biases are mitigated such that the geographical pattern and interactions between variables play more weight in the resulting lightning prediction.

CanESM's LCL matches MERRA-2 LCL well (Figures 3b, S3 and S4), except near the poles, which is discussed further below. CanESM's $r$, the column saturation fraction, is biased high (Figures 3c, S5 and S6) across most of the globe. The high bias in $r$ likely compensates for the low CAPE bias, resulting in an overall global lightning occurrence that is of reasonable magnitude. Note that both LCL and $r$ from CanESM5.1 drop greatly near the poles (>75°) when looking at the annual mean figures. During polar night, both LCL and $r$ drop close to zero, which implies that CanAM5.1's atmosphere in polar night is too cold and dry for adequate moisture and condensation. For context, we also plot the northern summertime (June-July-August) means in Figures S3 and S5 in order to show that when there is sunlight at the Arctic pole, the LCL and $r$ results are more reasonable there. While it is highly unlikely for lightning (and even more unlikely for natural fire ignition) to occur near the poles during polar night, we nevertheless keep this in mind when plotting the seasonal cycles and when interpreting the results in Section 4.1.



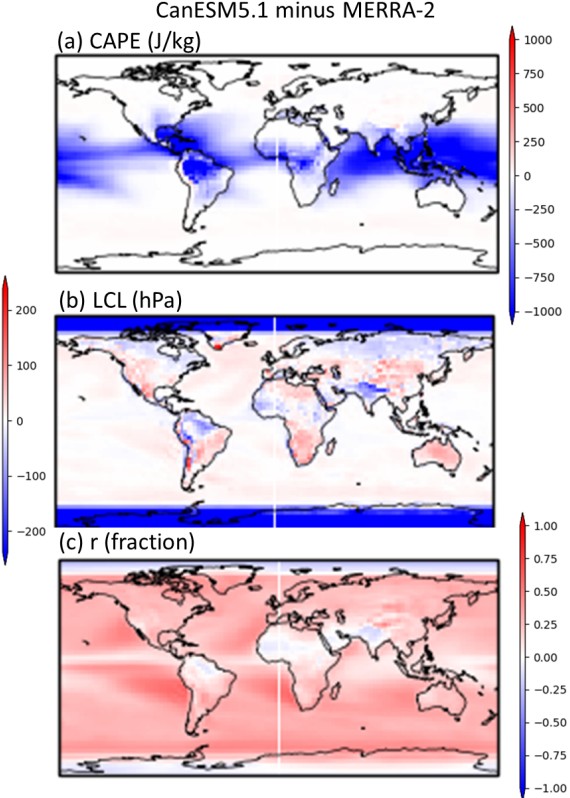

**Figure 3.** CanESM5.1 minus MERRA-2 differences for input parameters (a) CAPE, (b) LCL, and (c) r.

### 3.3 Lightning flash rate

As with lightning occurrence, the spatial distribution of lightning flash rate (Figure 4) is quite good compared to the LIS/OTD climatology. The annual global mean modelled lightning flash rate underestimates that from LIS/OTD by -1.5 flashes/km$^2$/year. The model underestimations are apparent in Figures 4c and 5a in most of the tropics and in the southeast United States. As cloud-to-ground flash rate gets used in CanESM's wild fire ignition, this means that burned area in those regions will likely be underestimated by CanESM5.1 when this lightning scheme is used. In the global mean bias, those underestimations are

balanced by the model overestimations that occur along the west coast of North and South America, and, to a lesser extent, over other high-latitude land regions and in the southern ocean.

The zonal mean shape in lightning flash rate from LIS/OTD (Figure 5a) is more symmetrical around the equator than the ISS LIS lightning occurrence (Figure 2a), with the larger difference in the southern hemisphere. The modelled zonal mean flash rate does not have the same peaks as LIS/OTD. While the LIS/OTD data product is global, it is important to note that

its results are more uncertain at high latitudes, where only OTD contributes to the data product. In addition, the global mean seasonal cycle of modelled flash rate is centered on May-June (Figure 5b), while the LIS/OTD flash rate peaks in August. This result is consistent with Figure 2b, which showed positive model biases in flash rate and an earlier seasonal peak compared to observations.





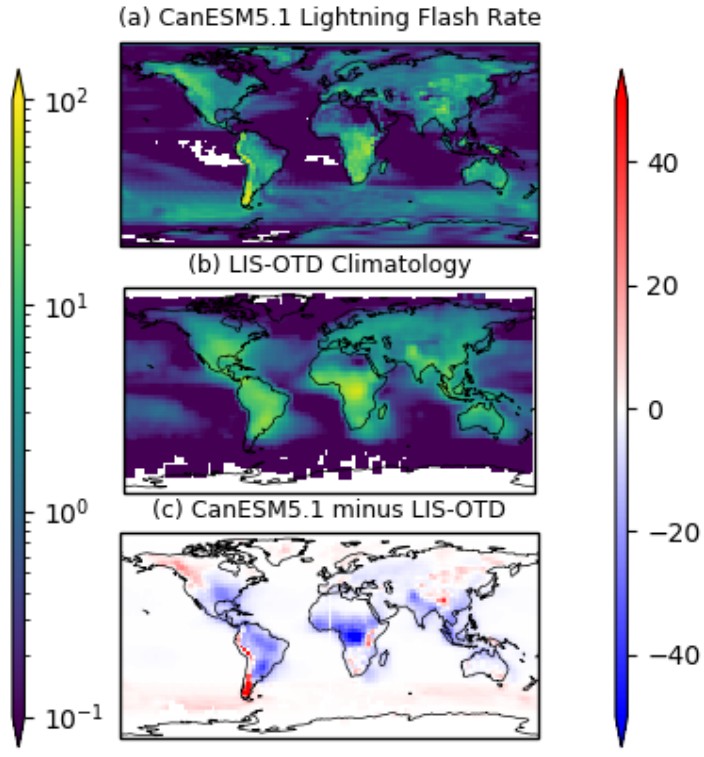

**Figure 4.** Comparison of the 1995-2014 mean modelled lightning flash rate to the LIS-OTD observed climatology, and their difference (all in flashes/km$^2$/year).

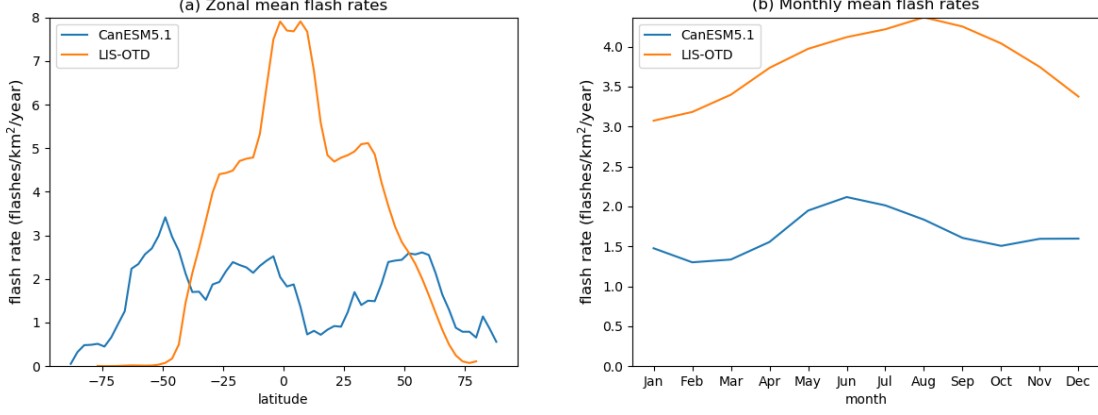

**Figure 5.** (a) Comparison of the zonal mean flash rates and (b) seasonal cycle from CanESM (model), and the LIS-OTD observed climatology.





Our lightning evaluation results are on par with evaluations of other lightning parameterizations. For example, several stud-
ies, such as Tost et al. (2007); Finney et al. (2014); Gordillo-Vázquez et al. (2019); Stolz et al. (2021) evaluated four, five,
six, and three different lightning parameterizations, respectively, and found that the predicted global total lightning is often 2-3
orders of magnitude off of that observed. Ours is within a factor of 2 (model over measurement mean of 0.5).

### 3.4  Burned area

Similar to lightning, the modelled burned area (BA, in Mkm$^2$) is dependent on its underlying variables, and biases in those
will cause biases in BA. However, we can examine the impact of lightning options by assessing modelled BA for two different
CanESM5.1 simulations: that with the Etten-Bohm et al. (2021) lightning parameterization ("interactive lightning"), and that
with the unchanging monthly LIS/OTD climatological lightning ("control lightning").

Both have the same human ignition source, with the resulting BA shown in Figure 6a and b, respectively, and are evaluated
against the MODIS Fire_cci burned area grid product, version 5.1 (Figure 6c). The mean global total BA for 2017-2019 is (a)
3.88 Mkm$^2$, (b) 7.18 Mkm$^2$, and (c) 4.16 Mkm$^2$. For additional context, Chuvieco et al. (2018) reported an average of 3.81
Mkm$^2$ for 2001 to 2016 for an earlier version of MODIS v5.0.

The bottom row of Figure 6 highlights the regional distribution of absolute differences between BA with interactive lightning
vs BA with control lightning (Figure 6d), as well as differences between modelled BA and the MODIS-derived BA (Figures 6e
and f). The global total absolute bias for the interactive lightning simulation is -0.28 Mkm$^2$ and that for the control lightning
simulation is +3.02 Mkm$^2$.

The spatial distribution of the model's over and underestimations of BA (Figure 6e) are only somewhat explained by the
spatial distribution in modelled lightning (Figure 4c). Other model biases related to BA, such as temperature and soil moisture
contribute as well to the differences. Indeed, using the climatological lightning, CanESM5.1 overestimates compared to the
MODIS-based BA (Figure 6f), with very high biases in the tropics due to known CanESM5.1 climate biases there (e.g., too
dry, resulting in too much combustion). The lower lightning in fire-prone areas from the interactive model have a compensating
effect for that.

The zonal mean BA from the interactive and control simulations and from MODISv5.1 are shown in Figure 7a. There we see
that CanESM5.1's BA with interactive lightning doesn't quite have the same latitudinal pattern. The BA with control lightning
is overestimated in the southern tropical region for climate bias reasons mentioned above. The positive bias in the tropics may
also be due to the cloud-to-ground fraction in the control lightning simulation, which is set to 0.25 globally. This fraction is
likely too high in the tropics (contributing to the large overestimate there), and too low at high latitudes. In newer versions of
the land model (currently only available offline), a latitudinal-varying cloud-to-ground fraction, similar to that of our interactive
lightning, is used. Both simulations agree better with MODIS from about 20-60°N, where the average cloud-to-ground fraction
is more applicable, and more consistent between the control and interactive lightning simulations.

Finally, the globally-summed monthly mean BA is plotted in Figure 7b to examine the seasonal cycles from the simulations
and MODIS. MODIS has peaks in August and December, whereas the control simulation has just one peak around October.

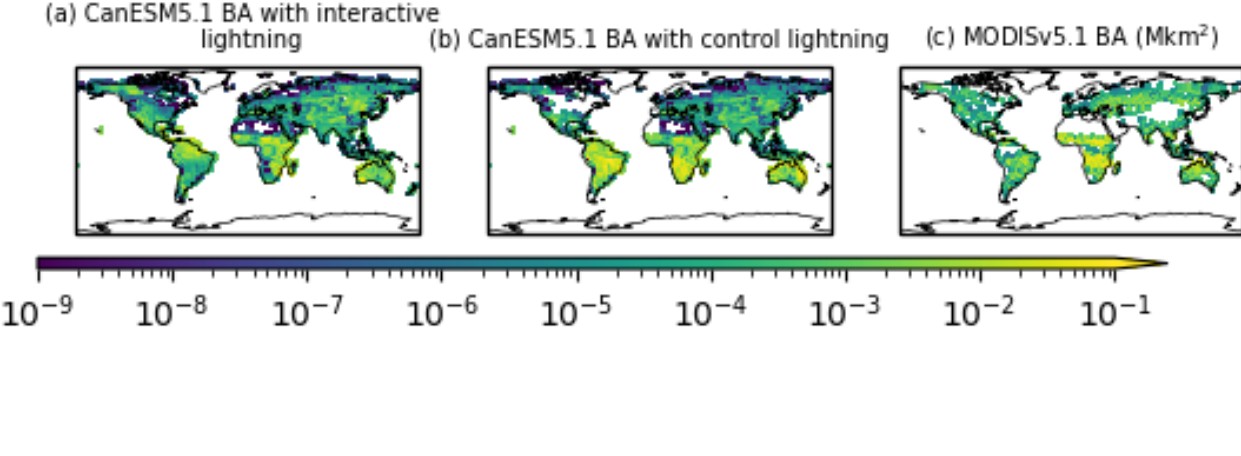

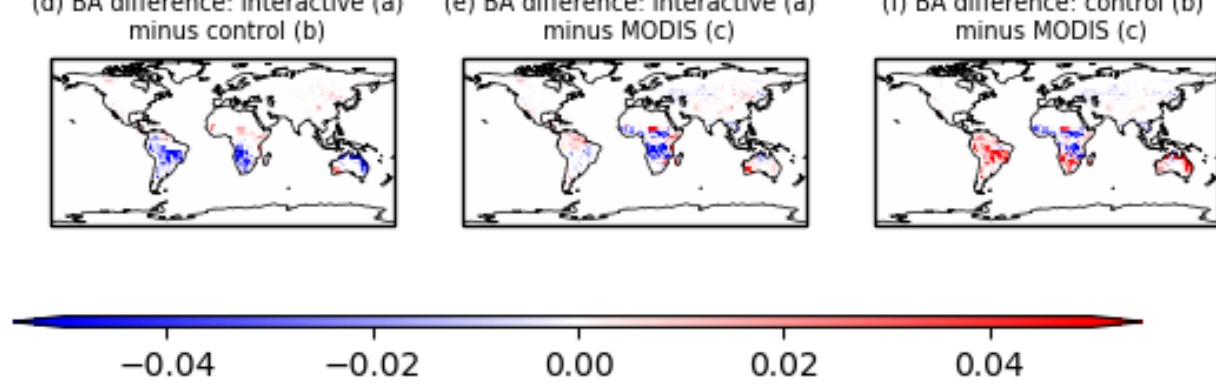

**Figure 6.** Comparison of the 2001-2014 annual mean burned area (BA), all in Mkm$^2$. BA from CanESM5.1 with (a) interactive lightning and (b) control lightning. (c) BA from MODIS v5.1. Note that the log color scale has no color when the value equals zero. Absolute difference between (d) panels a and b; (e) panels a and c; and (f) panels b and c.





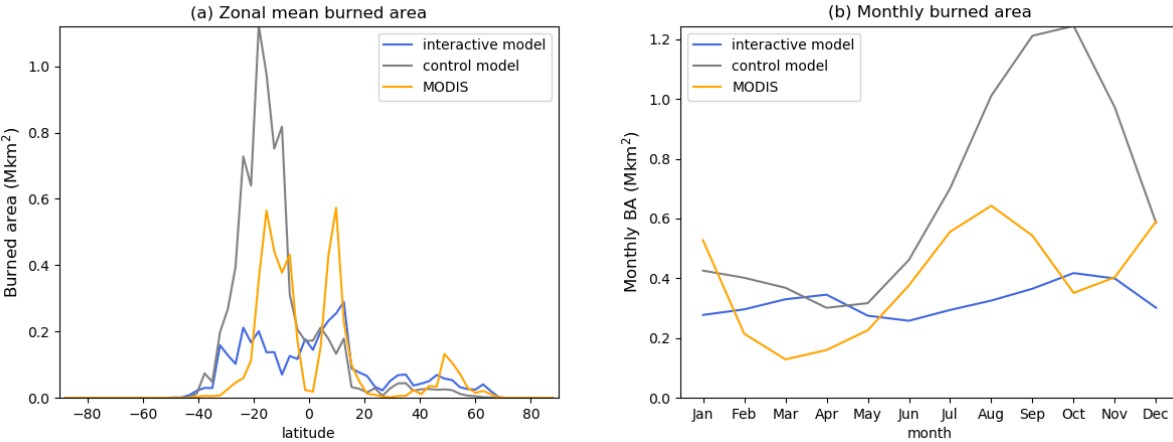

**Figure 7.** (a) Zonal mean and (b) monthly mean comparisons of the 2001-2014 annual mean burned area (BA).

The interactive simulation has small peaks in April and October/November, almost opposite the measured seasonal cycle. These differences in seasonal patters are likely related to the spatial model biases and local seasonality there.

CLASS-CTEM, CanESM5.1's land model, was evaluated along with several other vegetation models in Hantson et al. (2020), where they found that the global total simulated burnt area was within the range of GFED4's observational uncertainty of 3.45-4.68 $Mkm^2$ (GFED4 is based off of MODIS burnt area). The results from our study show that CanESM5.1 with interactive lightning results in comparable burnt area to those in Hantson et al. (2020).

## 4    Future projections of lightning and fire; 2015-2100

### 4.1    Future changes to lightning

After establishing satisfactory results for present-day lightning predictions, we now run 10-member ensembles of CanESM5.1 with the Etten-Bohm et al. (2021) lightning scheme into the future under the extreme climate change scenario, SSP5-8.5 and the more moderate scenario, SSP2-4.5. We average the last twenty years of the simulation ("future", 2081-2100) and subtract the result from the average of the "present" twenty years (2015-2035 average) to see how lightning changes in the future climate scenarios, and these results are shown in Figure 8. Both have similar spatial patterns and trends, but the moderate scenario has less a pronounced decrease in the tropics. Both scenarios have increases at mid-latitudes, particularly in Siberia and Western US. As noted in Section 3.1, results above 75°N are not reliable.

Our spatial trends in lightning under the higher emissions scenario are similar to those in Finney et al. (2018) and Etten-Bohm et al. (2024). The Finney et al. (2018) study used an upward ice flux lightning parameterization, and the Etten-Bohm et al. (2024) study used the same lightning parameterization as we do (Etten-Bohm et al., 2021), but applied in the CAM5 model. The difference in our lightning results and those in those two studies are due to (a) differences in the number of years

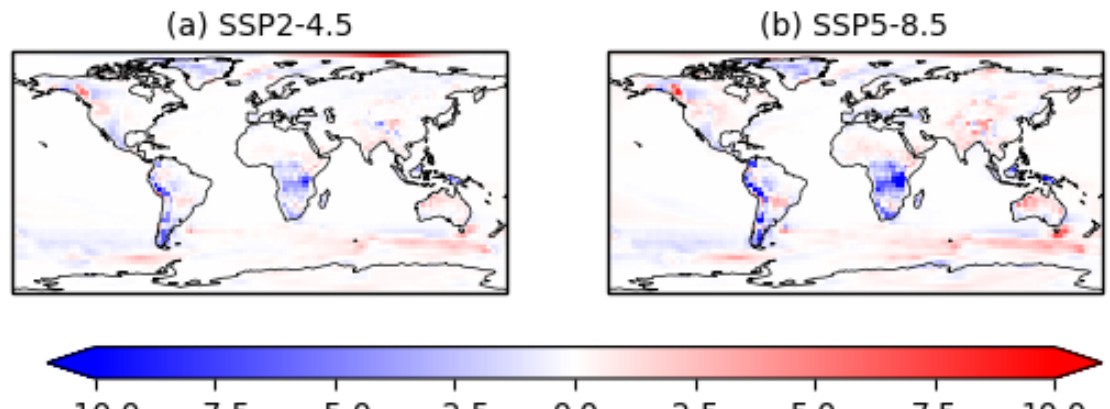

**Figure 8.** Absolute differences in lightning flash rate (flashes/km$^2$/yr): 2081-2100 average minus 2015-2035 for the SSP2-4.5 and the SSP5-8.5 scenarios.

**Table 2.** Global and regional percent differences in future (2081-2100) minus present (2015-2035) lightning flash rate for different climate scenarios.

| Region | Scenario | Percent change |
|--------|----------|----------------|
| global | SSP2-4.5 | -0.5% |
| N mid-lat | SSP2-4.5 | -0.3 |
| tropics | SSP2-4.5 | -4.7 |
| Arctic | SSP2-4.5 | -6.2 |
| global | SSP5-8.5 | 0.14% |
| N mid-lat | SSP5-8.5 | 4.5% |
| tropics | SSP5-8.5 | -9.7% |
| Arctic | SSP5-8.5 | -1.8% |

averaged in the future vs present, and (b) differences in the climate of the ESMs. All of these have impacts on future lightning projections.

When the regions are averaged, the difference in the last 20 years vs first 20 years are summarized in Table 2. Globally, we see a very small change ($<$1%. Clark et al. (2017) compared the global trends in lightning flash density through the end-of-the century for eight different lightning parameterizations implemented in CAM5 and found changes ranging from -6.7% to +45% for RCP-8.5, so our results fall within that range, but on the lower end.

The future relative changes to lightning are further highlighted in Figure 9, which shows the annual average lightning flash rate time series for the whole globe, the tropics (30°S-30°N), and the northern mid-latitudes (30-60°N) for both the SSP2-



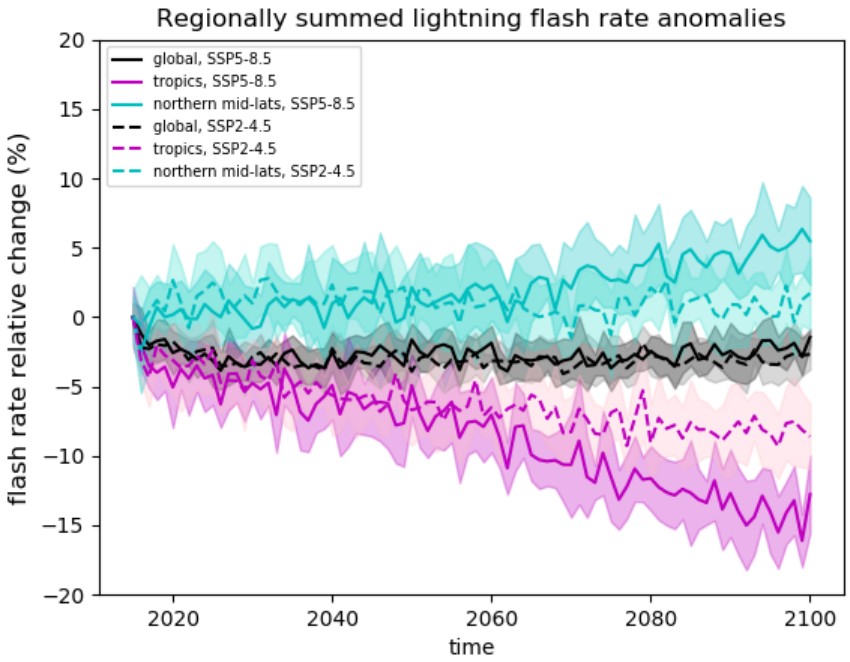

**Figure 9.** Regionally summed lightning flash rate anomalies compared to 2015 (in % change) over time for two future climate scenarios.

4.5 (dashed) and SSP5-8.5 scenarios (solid) as a percent change of each year compared to 2015. The shading represents the

280 standard deviation of the ensemble members. SSP2-4.5 trends are similar, but smaller in magnitude than those for SSP5-8.5.

Our results also indicate decreased lightning in the Arctic (60-75°N) by -1.8% (for 2081-2100 vs 2015-2035), with a large contribution from Greenland (Figure 8). The original lightning flashrate in the Arctic was very low and decreased rapidly in the first 15 years of our simulations, followed by a slow increase from the 2030 low point. Our overall decrease in Arctic lightning is in contrast to the conclusions by Chen et al. (2021). Chen et al. (2021) used a parameterization based on the product of

285 CAPE and surface precipitation rate to determine the lightning flash rate, and highlighted the threat of fires in the Arctic region due to the combination of increased lightning and vegetation. In their study, circumpolar region lightning increased by $112 \pm 38\%$ by the end of the century (2081-2100 with RCP8.5).

In the mid-latitudes, where lightning is increasing, we additionally examine the shift in seasonality that occurs in lightning occurrence (Figure 10). When comparing the last 3 years (2098-2100) and recent 3 years (2017-2019), we see that the lightning

season will start earlier in the year, and will increase late in the season as well. These shifts have implications for extending the Boreal forest fire season with severe climate change.





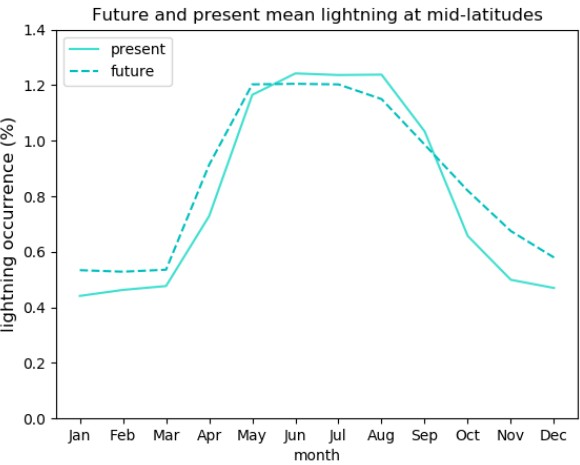

**Figure 10.** Mid-latitude seasonal mean in lightning occurrence (%) in the present (solid line, 2017-2019) and future (dashed line, 2098-2100) in SSP5-8.5.

## 4.2 Future changes to burned area

In some parts of the world, wildfires have increased in frequency and intensity due to climate change (Flannigan et al., 2005; Hope et al., 2016; Halofsky et al., 2020; Kirchmeier-Young et al., 2019). These increases are likely to continue as temperatures rise in the future. That said, future fire emissions in the CMIP6 project were actually projected to decrease globally due to land use changes (climate influences on fire emissions not taken into account). Therefore, we examine how BA will change in the future from CanESM5.1 simulations with and without the new interactive lightning parameterization. That is, we examine future BA ignited with evolving lightning from Section 4.1 versus a "lightning control run" that has the unchanging, monthly climatological lightning from present day (the LIS/OTD dataset) used throughout the 2015-2100 simulation. Both simulations have the same changing climate (temperature, precipitation, moisture, land use change, etc), and same unchanging human ignition. The differences between the two simulations are explored to see only the impact of online lightning ignition.

### 4.2.1 With interactive lightning

Figure 11 shows the future (2081-2100) minus present (1995-2014) annual mean BA (given as a percent of model grid cell burned) for the moderate and extreme future climate scenarios for the evolving lightning simulation. There are distinct regional differences in where BA increases or decreases. Globally, BA has a mean change of about +18% in the SSP5-8.5 scenario and +10% for SSP2-4.5.

These changes are further illustrated as a time series of anomalies in $km^2$ (first year subtracted off values) in Figure 12a. There is a large difference in the tropics and thus the global total between the SSP5-8.5 and the SSP2-4.5 scenario. For SSP2-4.5, the decreasing part of the global time series is likely due to the combination of reduced lightning in the tropics (Figure

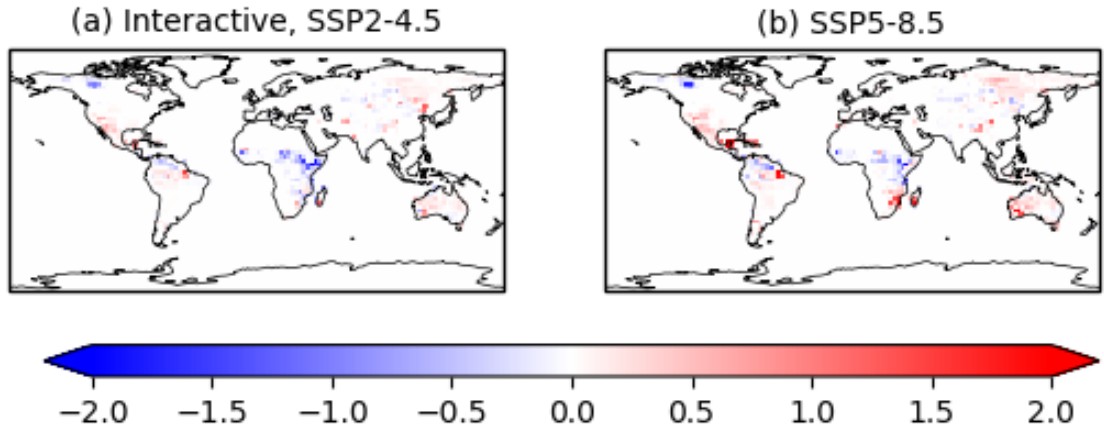

**Figure 11.** Future (2081-2100) minus present (1995-2014) absolute differences in burned area (% of model grid cell) for the SSP2-4.5 and the SSP5-8.5 scenarios - with interactive lightning.

8a), combined with less severe warming. In SSP5-8.5, the annual global total BA is 4.5 Mkm$^2$ in the future vs 3.8 Mkm$^2$ in the present. With evolving lighting, northern mid-latitude BA increases in both SSP2-4.5 and SSP5-8.5 scenarios by 22% and 36%, respectively.

### 4.2.2 With control lightning

Conversely, Figure 13 shows the geographical patterns in BA for constant, unchanging lightning. In this case, when comparing 315 the last 20 years to the first 20 years, the global total future annual mean BA for SSP5-8.5 (Figure 13b) is 10.8 Mkm$^2$ vs a much lower 7.5 Mkm$^2$ in the present, representing a large increase globally of +43%. For the SSP2-4.5 control lightning simulation (Figure 13a), there was a smaller increase in BA. Figure 12b shows that the difference between SSP5-8.5 and SSP2-4.5 happens mainly in from 2060 onward.

Note that with constant lightning, the change to Northern mid-latitude BA is much smaller (Figures 12b and 13), which is 320 in contrast to the case when lightning is evolving with climate. With unchanging (control) lighting, mid-latitude BA increases in both SSP2-4.5 and SSP5-8.5 scenarios by 7.5% and 13%, respectively. These are about a third the percent increase at mid-latitudes than the interactive lightning scheme had.



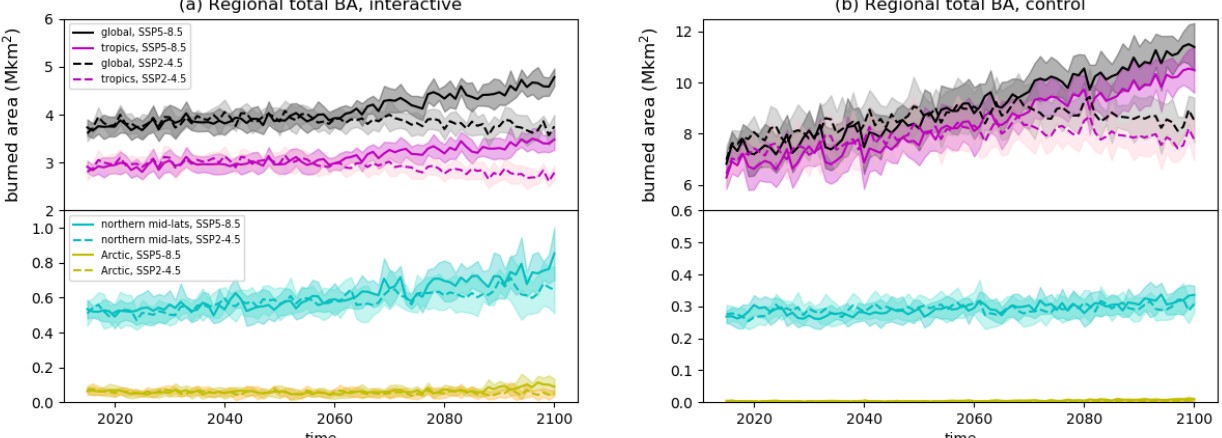

**Figure 12.** Time series of regional and global total BA (in Mha) for the SSP2-4.5 and the SSP5-8.5 scenarios - with (a) interactive lightning, and (b) control lightning. Note the different y-axis ranges.

**Table 3.** Global and regional percent differences in future (2081-2100) minus present (2015-2035) BA for different climate scenarios and different lightning.

| Region | Scenario | Control lightning | Interactive lightning |
|--------|----------|-------------------|-----------------------|
| global | SSP2-4.5 | 8.7% | -1.7% |
| N mid-lat | SSP2-4.5 | 7.5% | 23% |
| tropics | SSP2-4.5 | 7.9% | -7.0% |
| Arctic | SSP2-4.5 | 21% | -11.5% |
| global | SSP5-8.5 | 43% | 19% |
| N mid-lat | SSP5-8.5 | 13% | 36% |
| tropics | SSP5-8.5 | 42% | 14% |
| Arctic | SSP5-8.5 | 142% | 36% |

### 4.2.3 Future BA: interactive vs control lightning

There is a large fire difference between simulations that allow lightning to evolve with climate ("interactive lightning") versus

unchanging ("control") lightning, summarized in Table 3. When lightning is allowed to evolve, lightning will decrease in parts of the tropics (e.g., Figure 9), and thus global BA (which is dominated by the tropics) won't increase as much as when lightning is held static. For example, in SSP5-8.5, the global increase in BA is about a third as much with realistically evolving lightning (18% vs 54%). For SSP2-4.5, it is about half as much (10% vs 23%). Conversely, the northern mid-latitude BA is significantly greater with evolving lightning compared with constant lightning (Table 3).





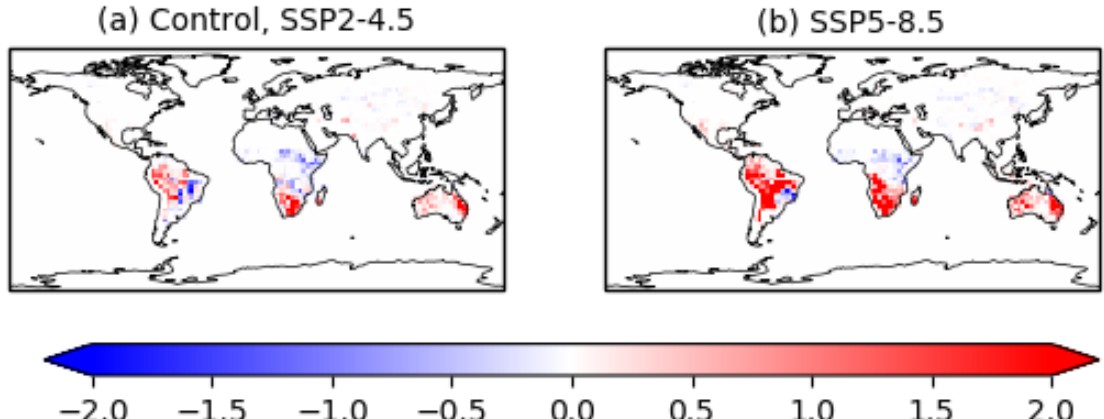

**Figure 13.** Future (2081-2100) minus present (2015-2035) absolute differences in burned area (in % of model grid cell) for the SSP2-4.5 and the SSP5-8.5 scenarios - with unchanging lightning.

One should expect these results to be heavily dependent on range of years averaged and the lightning scheme used. For example, a cloud-top height lightning scheme projects lightning increasing everywhere in the future climate (Finney et al., 2018), and would thus have much greater fires simulated in the future.

## 5 Conclusions

This study represents the first time a lightning scheme was implemented in CanESM5.1 and the first time the CLASS/CTEM fire scheme was driven interactively with CanESM's atmospheric physics. The logistic regression lightning "model b" from Etten-Bohm et al. (2021) was used, as it is calculated from model environmental variables (i.e., CAPE, LCL, and $r$) that we have higher confidence in compared to already parameterized cloud and precipitation variables. The lightning model is just one equation that applies everywhere globally and does not require tuning. To our knowledge, CanESM5.1 is also only the second model to apply the Etten-Bohm et al. (2021) lightning parameterization (the other being CAM5 in Etten-Bohm et al. (2024)).

The lightning occurrence and flash rate from CanESM5.1 were evaluated against satellite measurements, and the model produced realistic lightning spatial distribution and magnitude, with an exceptionally good land/ocean ratio. Overestimations still occurred in mountainous regions despite removing the main LCL term from the lightning calculation to improve results. While there is little-to-no lightning observed in the high Arctic, our analysis of the input variables LCL and $r$ indicate that CanESM5.1 does not have reliable results above 75° latitude.

When simulations were run out to 2100, we found that the future climate (SSP5-8.5) caused global total lightning to only change by 0.14%. However, there is a clear decrease in lightning in the tropics (-10%) and increase at mid-latitudes (+5%). The latter includes the boreal forest region (40-60°N), which is becoming more susceptible to wildland fires. These changes were similar, but smaller, with the SSP2-4.5 scenario.

The applications of this new lightning scheme in CanESM5.1 allowed for a more interactive and variable distribution of
burned area. Our simulations with online "evolving" lightning showed regional increases and decreases in burned area, and
those results were compared to a future simulation that had offline, unchanging, lightning. The results were significantly
different from one another (by up to a factor of 3 at mid-latitudes), showing the importance of (a) having a realistic lightning
scheme that will respond appropriately to changing climate, and (b) that lightning ignition is an important climatological factor
for future fire simulations, in addition to changes in temperature, precipitation, moisture, etc.

The application of this new lightning scheme in CanESM5.1 has given us the ability to have lightning changing online
with CanESM climate, as well as the capability to better model tropospheric $O_3$ in future work. We recommend this online
lightning scheme to continue to be used in CanAM and CanESM simulations that require comprehensive SLCF simulations.
Future work will include evaluating this lightning scheme at higher resolution ( 1°) with CanAM's new dynamical core, GEM,
which utilizes a yinyang grid - having no singularity at the poles. One would expect better $r$ and LCL near the poles in CanAM-
GEM, and thus, high-Arctic results may be more reliable in that version. One would also expect the lightning scheme to work
well at higher resolution since it was trained on data with 0.5° resolution (Etten-Bohm et al., 2021). We would also like to
evaluate and apply this lightning scheme at even higher resolution in the regional climate model, CanRCM (Scinocca et al.,
2016), over a North American domain.

*Code and data availability.*   The LIS/OTD lightning climatology dataset can be found online, here: https://ghrc.nsstc.nasa.gov/uso/ds_docs/
lis_climatology/LISOTD_climatology_dataset.html with additional information here: https://ghrc.nsstc.nasa.gov/uso/ds_details/collections/
loCv2.3.2015.html

The ISS LIS datasets are available online here: https://ghrc.nsstc.nasa.gov/lightning/data/data_lis_iss.html

The MERRA-2 datasets are available online here: https://gmao.gsfc.nasa.gov/reanalysis/MERRA-2/

And the MODIS fire_cci burned area grid product, v5.1 is available online here: https://catalogue.ceda.ac.uk/uuid/3628cb2fdba443588155e15dee8e5352
The CanESM5.1 model code is available online, here: https://gitlab.com/cccma.

*Author contributions.*   CHW put the lightning scheme into CanESM5.1, did the model runs, wrote the paper, and created the figures. MEB
advised on the lightning scheme and processed the ISS LIS and MERRA-2 datasets. CS advised on the lightning scheme and provided
scientific guidance for this project. VA developed CanESM's land model, including its fire model and provided modelling support. AA, JC,
ML, DP, and KvS provided CanESM modelling support and guidance.

*Competing interests.*   The authors have no competing interests.

*Acknowledgements.*   This work was in part supported by NASA Grant NNX17AH66G to Texas A&M University.



The LIS/OTD climatological dataset is provided through the NASA Earth Science Data and Information System (ESDIS) Project and the Global Hydrology Resource Center (GHRC) Distributed Active Archive Center (DAAC). GHRC DAAC is one of NASA's Earth Observing System Data and Information System (EOSDIS) data centers that are part of the ESDIS project.





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
