# Peer review of "A new lightning scheme in Canada's Atmospheric Model, CanAM5.1: Implementation, evaluation, and projections of lightning and fire in future climates"

_Geoscientific Model Development, 2024_

## Author Comment (AC1)

(b) LFD Trend During 1996-2013

$10^{-1}$ fl km$^{-2}$ year$^{-1}$

**Fig AR1(a):** Observed lightning trends from LIS from 1996-2013, as shown in Qie et al. "Regional trends of lightning activity in the tropics and subtropics", Atmos. Res., 242(11), 104960, 2020.

**CanESM5**

- SSP5-8.5 represents the high end of the range of future pathways, corresponding to RCP8.5
- 2100 avg vs 2015 avg

**Finney et al (2018) IFLUX**

Change in flash rate 2100-2000 under RCP8.5 (fl. km$^{-2}$ yr$^{-1}$)

- UK model with IFLUX (upward cloud ice flux lightning scheme), also RCP8.5
- 2100 vs 2000 avg

**Etten-Bohm et al (2024) CAM5**

- Same lightning, but in CAM5
- Lightning occurrence (%)
- 2098-2100 vs 2017-2019 avg

**Fig AR1(b):** future minus present changes in lightning: our study, vs two other studies.

---

## Author Comment (AC2)

[Figure]

**Fig 1:** (left) CanAM simulation with adaptive entrainment and detrainment deep convection scheme for a 1-year simulation for 2003. (right) CanESM simulation with default deep convection scheme for a 3-year simulation for 2017-2019 (Fig S1 in original manuscript's supplement).

[Figure]

**Fig 2:** Zonal evaluation of lightning flash rates. (left) Figure 5a from the original manuscript, (right) after the CAPE standardisation correction.

---

## Author Response (AR1)

Reviewer comments in *italic*
Authors responses begin with [AR]

**Review 1:**
*General comments:*
*There were no lightning schemes (no lightning calculated) in Canada's Earth System Model 5.1 before this study, which introduced a lightning scheme into CanESM5.1. The Etten-Bohm scheme is based on a logistic relation between lightning occurrence and multiple environmental variables (CAPE, LCL, r), which is unique among existing lightning schemes. The Etten-Bohm scheme is expected to show a relatively acceptable performance in CanESM, but unfortunately, due to the large biases in CAPE, LCL, and r simulations in CanESM, the lightning simulation accuracy cannot be considered satisfactory.*

[AR] Thank you for your thorough review of the paper. As will be discussed below, we found and corrected an error in the lightning implementation in CanAM that has resulted in improved performance in our revised manuscript. The remaining model/measurement differences are smaller, and well within the range of performance of other lightning parameterizations in ESMs. Further, the revised lightning results sufficiently capture the latitudinal and seasonal variations necessary to assess wildfire burned area in present day and future climate scenarios simulated by CanESM. Our results continue to show an excellent land/ocean discrimination of lightning occurrence.

We've also added more explanation in the revised manuscript (Section 2.1) that, while CanESM has some differences in basic state parameters compared to MERRA-2, since the CAPE, LCL, and r inputs are standardised (e.g., (CAPE - meanCAPE)/standard_dev(CAPE)) before being input to the lightning calculation, the influence of their systematic bias is minimised, which is why the lightning results have a small error compared to ISS LIS and LIS OTD despite a systematic underestimate of CAPE and overestimate of r. This was mentioned in Section 3.2 of the original manuscript, but we have clarified that approach in the revised manuscript in the methods section (Sec 2.1). We note that all ESMs will experience basic state and cloud variable biases compared to the real atmosphere, but that the standardisation minimises these issues, motivating the use of this parameterization.

*I am not sure about the statistical metrics such as R and RMSE (root mean squared error) values between lightning simulations and observations in this study, but previous studies achieved better lightning prediction accuracy in their models (Finney et al., 2014; Lopez, 2016; He et al., 2022).*

[AR] In terms of the accuracy of lightning prediction compared to other parameterizations, Fig. 1.1 (from Clark et al. 2017) shows that other lightning parameterizations implemented in CAM5 experience a wide range of predictions, with a notable struggle in representing a realistic land/ocean contrast. Finney et al. (2014) further show a wide range in statistical accuracy when a suite of lightning parameterizations were applied to ERA-I data. In the revision, we have added RMSE and R statistical metrics to the evaluation (Section 3.1 and 3.3), and discuss them relative to other studies so that the reader can better evaluate the performance of the parameterization.

[Figure]

Fig.1.1: A diverse set of lightning parameterizations utilised in CAM5 from Clark et al. (2017).

*Is this paper valuable for publication? Yes, this is the first time the authors introduced a new lightning scheme to CanESM. However, I considered the following aspects that may improve the scientific significance of this paper:*

1. *If possible, try to implement more existing lightning schemes (as you listed in L35-L45) into CanAm5.1 and compare those lightning schemes with the Etten-Bohm scheme. The benefit of doing so is that you can evaluate how different lightning schemes influence the prediction accuracy to what degree. By comparing several different lightning schemes in a New ESM, you can provide more information for the scientific community.*

[AR] Thank you for this suggestion. While it would be a valuable exercise, it was not the purpose or in the scope of this paper to test several lightning schemes in CanESM. The main goal of our study was to test the implementation of the Etten-Bohm et al (2021) parameterization and show how including interactive lightning affects burned area in a changing climate compared to assuming unchanging climatological lightning. We note that the Clark et al. (2017) study mentioned above, as well as Gordillo-Vazquez et al (2019) have evaluations of multiple lightning parameterizations in CAM5, and Etten-Bohm (2024, currently in review in GRL, [link to preprint](link)) compares the implementation of the parameterization to the Romps et al. (2014) CAPE*Precip parameterization in CAM5. Therefore, that information is already available for the scientific community.

*…I also recommend that you consider the uncertainties that existed in observations when evaluating lightning schemes (you can refer to (He et al., 2022) Figure 4). Even you can use a second ESM (such as CESM) to provide additional supporting information (for example, if CESM can better represent CAPE, LCL, how these improvements can influence the prediction accuracy of lightning).*

[AR] Thank you for the He et al. (2022) reference, we have added information about the uncertainties of the observations into the revised manuscript (Sections 2.2.3 and 3.3). In terms of using a second ESM for supporting information related to basic state biases, this is done in Etten-Bohm et al. (GRL, *in review*) who used the same lightning scheme we have, but in CAM5. We discussed the similarities and differences between our results and theirs in our paper (Section 3.1 and 4.1, was on lines 158, 161, 267-270 of the original manuscript).

We have also done further tests in CanESM5 using a TKE boundary layer scheme as well as an alternate deep convection scheme called adaptive entrainment and detrainment (AED) by

McFarlane, Scinocca, & Abraham (as yet, not published). The former did not have a significant impact on CAPE, but the latter had a great improvement to CanESM's CAPE (Fig. 1.2 below).

[Figure]

**Fig. 1.2:** (left) CanAM simulation with AED deep convection scheme for a 1-year simulation for 2003, showing marked improvement over, (right) CanESM simulation with default deep convection scheme for a 3-year simulation for 2017-2019 (Fig S1 in original manuscript's supplement). Note the different colour scales at far left over each figure.

CanAM's default deep convection scheme is the Zhang McFarlane (ZM) scheme (more on that below). While the AED scheme improved CAPE, the lightning results from both ZM and AED were very similar due to the standardisation of input variables that occurs before lightning is calculated.

Nevertheless, in our investigation of this, a latitudinal-based error was discovered and rectified in the CAPE standardisation step of the new lightning subroutine, and that fix has greatly improved the zonal distribution of lightning occurrence and lightning flash rate (for example, Fig. 1.3 below, and Figs 2a and 4a in the revised manuscript). Therefore, in our revised manuscript, we have redone all simulations with this important correction, and it has resulted in a better global lightning distribution. Thank you for flagging this in your review!

[Figure]

**Fig 1.3:** Zonal evaluation of lightning flash rates. (left) Figure 5a from the original manuscript, (right) after the CAPE standardisation correction (Fig 4a in revised manuscript).

*2.      I am not sure whether if you applied meteorological nudging (u, v, T) in your simulations, by the way, you need a detailed experiment setup in your paper (use a chart). You can evaluate how meteorological nudging influences the prediction accuracy of lightning in the Etten-Bohm scheme by turning on/off nudging. For example, if meteorological nudging can significantly improve the simulation accuracy of CAPE et al. as well as lightning, this can prove that improving the simulation of CAPE et al. in CanESM can definitely lead to the improvement of lightning simulations (and the extent of this improvement to which degree).*

[AR] The simulations were not nudged. In CanESM5, nudging does not directly target the variables that are of most interest (i.e., we do not nudge CAPE, LCL, or r), and we typically find that nudging degrades cloud properties and precipitation in the model, even though temperatures, winds, and humidity are improved. This is because cloud and convection parameterizations have been developed and tuned using observational constraints using un-nudged simulations. Typically, the skill of convection parameterizations in global models is assessed through comparisons of precipitation patterns. Given that we standardised the CAPE, LCL, and r input to the lightning equation, in the revision, we de-emphasize the comparisons of CAPE, LCL, and r by keeping figures of them only in the supplementary materials, and we clarify our approach that standardises the input variables and optimises lightning flash rates. Also note in our response above that we corrected an error in the CAPE standardisation that improved the lightning results. We also confirmed with an alternate deep convection scheme that while CAPE can be improved, its impact on lightning was minimal.
In the revision, we add more detail to the experiment setup (Section 2.2) for clarity.

*3.      What caused the large biases in CAPE, LCL, r? → Please provide some explanations in your paper. You mentioned you will use a new TKE deep convection scheme, if possible, please test this new convection scheme and estimate how it impacts the simulations of lightning and burned area.*

[AR] Firstly, we should correct that the TKE scheme is actually for the boundary layer, and not deep convection. Therefore, we have removed its mention from manuscript in the revision. That said, we did test it, and found it had no impact on CAPE or lightning.
CanESM's LCL doesn't have a large bias. And while CAPE is biased low, our tests did find that it is greatly improved with the AED convection scheme, which is as yet unpublished, and untuned in CanAM. We have added some text in this regard to Section 3.2.
Both the TKE boundary layer scheme, and the AED deep convection scheme will only be available in CanAM5.2, and thus were not included in this paper, for which we used the most recent version available (CanAM5.1).
Again, since each input variable is standardised before going into the lightning calculation, their systematic biases do not greatly impact the lightning occurrence results -- it is only their regional biases that do.

*4.      You conducted future projections; however, I recommend that you can firstly evaluate the response of global lightning activities to short-term surface warming (1993-2013) of Etten-Bohm scheme in CanESM. Evidence shows that there was no statistically significant trend in global lightning activities during 1993-2013 (LIS/OTD, (Williams, 1992)).*

[AR] Thank you for the suggestion to provide an evaluation of the 1993-2013 time series. This has been assessed for the revised paper (Section 3.3). Note that Qie et al. (2020, Atmos. Res.) use LIS observations to show statistically significant decreasing trends from 1996-2013 in parts

of the tropics (Fig. 1.4a below), and those spatial patterns in lightning trends are consistent with what we simulate for 1995-2014 (Fig 1.4b).

[Figure]

**Fig 1.4(a):** Observed lightning trends from LIS from 1996-2013, as shown in Qie et al. "Regional trends of lightning activity in the tropics and subtropics", Atmos. Res., 242(11), 104960, 2020.

[Figure]

**Fig 1.4(b):** Simulated lightning trend from 1995-2014 from CanESM historical ensemble (2012-2014 3-year mean minus 1995-1997 3-year mean).

[Figure]

- SSP5-8.5 represents the high end of the range of future pathways, corresponding to RCP8.5
- 2100 avg vs 2015 avg

- UK model with IFLUX (upward cloud ice flux lightning scheme), also RCP8.5
- 2100 vs 2000 avg

- Same lightning, but in CAM5
- Lightning occurrence (%)
- 2098-2100 vs 2017-2019 avg

**Fig 1.4(c)**: future minus present changes in lightning: our study, vs two other studies.

Also note that for 2015-2100 (Fig. 1.4c, left). Our overall future lightning trends also show general agreement with those from the sophisticated ice flux scheme in Finney et al. (2018), and that in Etten-Bohm et al. (in review) using the same lightning scheme in CAM5 (Fig. 1.4c, middle and right, respectively). This type of future trend evaluation is important for the goals of

this paper (see also response to reviewer 2).

*5.       You predicted a decreasing lightning trend with global warming (Table2), what is the reason (CAPE, LCL, r decreased?)? He and Sudo (2023) suggested that historical global warming enhanced lightning activities, but increases in aerosol burden exerted an opposite effect (1960-2014). Can you also separate the effects of warming and aerosols?*

[AR] In table 2, the global total change in lightning was very small (+0.8% in SSP5-8.5 and -0.1% in SSP2-4.5). Regionally, both warming scenarios had decreasing lightning, dominated by the tropics, though SSP5-8.5 had increasing lightning in the northern mid-latitudes. In order to better understand their relation to CAPE, LCL, and r, in the revision, we include the future minus present difference in each of the input parameters to more clearly show what changes are causing the lightning trends (Fig S7, S8, and S9 in revised Supplement, and also in Fig. 1.5 below). There we see that both CAPE and r contribute to the decreasing lightning trend in the tropics, although the interaction terms also need to be considered.
Our lightning scheme inherently includes changes from temperature and bulk aerosols, in how they impact climate and thus, CAPE, LCL, and r, but have not been teased out, as this lightning scheme does not depend on them directly.

[Figure]

**Fig 1.5:** Future minus present changes to LCL, r, and CAPE for SSP5-8.5 [red colour would contribute to an increase in lightning and blue colour would contribute to a decrease in lightning. Note the opposite colour bar of LCL (in pressure coordinates) to account for this].

*6.      "Control lightning" vs. "interactive lightning", what is the implication? The simulated lightning trends can largely impact the simulated burned area, but this is not a new finding.*

[AR] The influence of lightning on burned area is a significant finding since lightning is only one input for burned area and they are not linearly related. For example, lightning could strike, but if the fuel is very wet, or winds are very low, then burned area could remain minimal. There are several other factors for burned area, such as: proximity to human populations for ignition and suppression, the availability of biomass, the moisture of the biomass, which changes with precipitation, as well as temperature and winds for fire growth (humidity, vegetation type, etc as well). All of these impact burned area, and will have different relative importance regionally. Thus, the information added from providing burned area results from (a) an unchanging climatological lightning vs (b) interactive lightning is important new quantitative information that isolates the impact of lightning from the many other factors.
Here, we show that with our configuration, the relative importance of lightning is about as great as the difference in climate between the SSP245 and SSP585 scenarios. This has not been shown in the literature to our knowledge.

*Anyway, please try to improve the scientific significance.*

[AR] The scientific significance of our study is that we used a new lightning scheme that doesn't depend on highly uncertain cloud or precipitation variables in our climate model and got good results, including excellent ocean-land lightning gradients, and a similar response to climate warming as a process-based ice flux lightning scheme. We also quantify and emphasise the significant importance that lightning has on burned area now and in the future, which is not intuitive, given that fire depends on several other factors, such as fire weather, which is more prominent in the literature. The significance of our study has been emphasised in our responses above, in the original manuscript, and further highlighted in the conclusions.

*Specific comments:*
   *1. Please show statistical metrics (R, RMSE, MBE) between simulations and observations.*

[AR] We have added R and RMSE evaluation metrics to the revised manuscript (sections 3.1 and 3.3).

*2.      He et al. (2022) recently developed a new lightning scheme based on Lopez (2016) and McCaul et al. (2009). This paper can provide additional information for your paper's introduction (L32, L35-L45, L48-49).*

[AR] Thank you. We now include discussion of He et al. (2022) in the revised manuscript (e.g. in the introduction, and in Section 3.3).

*3.      L104, Uman (1986) only mentioned a blurry concept, could you please provide the detailed equations (to calculate the fraction) and relevant explanations. Another widely used equation for calculating cloud-to-ground fraction was proposed by Princ and Rind 1993. I am not sure which one is better but for your reference.*

[AR] The cloud-to-ground fraction was set to a linearly increasing value based on latitude, with 10% fraction at the equator, and increasing towards the poles. The equation we've used is:
cgfrac = A*abs(lat)+B, where A=0.3/45, and B=0.1, to result in 10% at 0 degrees latitude and

40% at +/-45 degrees lat. We've corrected/clarified this in the text and added that Price and Rind (1993) has an alternate option.

*4.       Which cumulus convection scheme is used? You need to add a detailed experiment setup into your paper.*

[AR] CanAM5.1 has separate parameterizations for deep and shallow convection. They are described in von Salzen et al (2013). Both parameterizations of convection use the same input profiles of temperature, moisture, and chemical tracer mixing ratios, which were output from the prognostic cloud scheme rendering them statically stable and at most fully saturated. Both schemes are permitted to be active in the same grid cells at any time within specific physical constraints for each scheme (von Salzen et al., 2005; Xie et al., 2002). The cumulus parameterization of Zhang and McFarlane (1995) is used to represent the effects of deep convection (hereafter denoted by ZM) in the model. The ZM-parameterization is a bulk mass flux scheme which includes a representation of convective scale motions. Effects of shallow convection are parameterized following von Salzen and McFarlane (2002) and von Salzen et al. (2005). In the parameterization, parcels of air are lifted from the planetary boundary layer (PBL) into the layer above the PBL. Shallow cumulus clouds are formed once the parcels reach the level of free convection (LFC), at which the parcels become positively buoyant. Above the LFC the parcels are modified by entrainment of environmental air into the ascending top of the cloud and also by organized entrainment at the lateral boundaries of the cloud. The cloud-top mixing produces horizontal inhomogeneities in cloud properties and vertical fluxes which are parameterized using joint probability density distributions of total water and moist static energy. The initial growth phase of the cumulus cloud is assumed to be terminated when its top reaches its maximum level. The growth phase is followed by instantaneous decay, with complete detrainment of cloudy air into the environment. <– We have added this description to the revised Supplement (Text S0).

References:
von Salzen, K., Scinocca, J. F., McFarlane, N. A., Li, J., Cole, J. N. S., Plummer, D., … Solheim, L. (2013). The Canadian Fourth Generation Atmospheric Global Climate Model (CanAM4). Part I: Representation of Physical Processes. Atmosphere-Ocean, 51(1), 104–125.

von Salzen , K. , McFarlane , N. A. and Lazare , M. (2005). The role of shallow convection in the water and energy cycles of the atmosphere . Climate Dynamics , 25 : 671 – 688.

Xie , S. , Xu , K.-M. , Cederwall , R. T. , Bechtold , P. , Del Genio , A. D. , Klein , … Zhang , G. J. and Zhang , M. (2002). Intercomparison and evaluation of cumulus parametrizations under summertime midlatitude continental conditions . Quarterly Journal of the Royal Meteorological Society , 128 : 1095 – 1136.

Zhang , G. J. and McFarlane , N. A. (1995). Sensitivity of climate simulations to the parameterization of cumulus convection in the CCC-GCM . Atmosphere-Ocean , 3 : 407 – 446.

*5.       L159 over some parts of the western …*

[AR] Text revised.

*6. L174, But from about 30S to 50N, the zonal pattern is modelled correctly → The lightning is systematically underestimated in the model? Which parameters mostly contributed to this systematic bias?*

[AR] As mentioned above, the lightning has been improved in the revised manuscript. We have also added more discussion of the parameters that have contributed to the remaining model biases.

*7. L171, what does TKE represent?*

[AR] Turbulent Kinetic Energy, however, since the TKE boundary layer scheme has little to no impact on CAPE or lightning, we removed it from the revised manuscript. We also note in the revision (Section 3.2) that the alternate AED deep convection scheme actually plays a bigger role in improving CAPE structure globally (Fig 1.2 above).

*8. L198, Figure S3 and Figure3, it looks like there is a rectangular dark blue box over polar regions, which is weird. It looks like there are bugs in your computer program, please check and justify this situation.*

[AR] Thank you for catching this. It has been corrected in the revised manuscript/supplement.

*9. Figure 6 and Figure 11. The figures are blurry, please provide figures with at least 300 dpi.*

[AR] Thank you, we have recreated these figures with dpi=300 or greater, and they are less blurry now.

*10. Figure S2, model largely underestimated CAPE within low-latitude regions, could you please explain it?*

[AR] CAPE in CanAM with the Zhane McFarlane deep convection scheme is "reversible" CAPE, meaning for the reversible calculation all of the liquid water that develops as the parcel is lifted is carried along with the parcel. That reduces the magnitude of the parcel virtual temperature (density temperature) and therefore the magnitude of the buoyancy. An alternative, alternate entrainment detrainment (AED), deep convection scheme will be available in an upcoming version of CanAM5.2, and we ran some tests with that version. CAPE was significantly closer to MERRA2 CAPE [e.g. Figure 1.2(left) above]. However, the impacts to lightning were minimal, because the standardization of CAPE before input to the lightning equation removes systematic biases. Therefore, since the AED deep convection scheme has not yet been tuned, and CanAM5.2 not yet released, and the lightning scheme functioned well in both cases, we have proceeded in the revision with the original deep convection scheme, and add more discussion to the revised manuscript to explain these results.

*11. Figure S5, model can partially capture the spatial pattern of r but systematically overestimate r compared to MERRA-2, what is the reason?*

[AR] As with CAPE, this systematic model bias in r has minimal impact on the lightning results.

*Technical corrections*

*Supplement, Figure S0, in the following sentence:*
*uppermost level (zt) may be reached by moists onvection.*
*Change "onvection" to "convection".*

[AR] Thank you - this typo has been corrected in the revision.

**Review 2:**
*The manuscript presents some results of the CanAM5.1 model with a new lightning scheme. In general, the manuscript is interesting, but I think it does not fit the scope of GMD. [...] I think that the manuscript should be transferred to the more relevant journal (I think ACP is a good alternative).*

[AR] Thank you for your review. The description of a GMD "Development and technical paper" is given here: https://www.geoscientific-model-development.net/about/manuscript_types.html#item2 and it includes "new parameterisations for processes represented in modules. [...] usually include a significant amount of evaluation against standard benchmarks, observations, and/or other model output as appropriate." We chose this because we were introducing a new parameterization in CanAM and evaluating it. However, if the editor agrees with reviewer #2 that ACP is a more appropriate journal for this manuscript, then we will discuss that option with them.

*Judging from the title, a new lightning scheme is the main subject of the manuscript, but it is taken from previously published paper and described in just less than 80 lines.*

[AR] In the revision, we add additional information to the lightning description, which is mainly how we have standardised the 3 input variables. However, the description is relatively small because the parameterization itself is only one equation, and is fully described in its own paper (Etten-Bohm et al, 2021). Our paper is focused not on the development of that parameterization, but rather on its implementation and impacts of it in CanAM (the lightning scheme is "new" to CanAM).

*The rest of the manuscript is devoted to the analysis and evaluation of the CanAM5.1 performance in the modeling of the cloud related quantities, which fully define rather low (see figure 1, 2, 4, 5) accuracy of the simulated lightning frequency. [...] I understand that the accuracy of the proposed parameterization is comparable to other available parameterizations, but it is not shown in the manuscript.*

[AR] In the revision, the accuracy of the simulated lightning frequency is improved due to finding and fixing an error on the CAPE standardisation step in the model code (see response to reviewer 1 about model performance). We've also added RMSE and R results to the evaluation and discuss how our lightning results compare to those from other studies.

*Analysis of the burning areas and future projections looks interesting but irrelevant to the main aim of the manuscript.*

[AR] As mentioned in our introduction, there are two main reasons why ESMs/climate models need to have a good lightning parameterization: tropospheric ozone and fire ignition. ESMs in general are used for climate projections forward in time. While that first subject (tropospheric ozone) is out of scope of our paper, we felt it very important to assess whether the new lightning

parameterization still provided realistic fire ignition by looking at how it impacted the burned area.

Also, to only evaluate the historical period could lead someone to believe that the Price and Rind cloud-top-height lightning scheme is sufficient [e.g. Fig. 2.1(left) below, from Clark et al, 2017], when Finney et al (2018) showed that that scheme would result in only-increasing lightning worldwide in a warming climate [Fig. 2.1(right)], which is not accurate. Both, the observations from the last 20 years [Fig. 2.2(left)], as well as a lightning-process-based ice flux scheme [Fig. 2.2(right)] show that a warming climate results in a *decrease* in lightning in the tropics and increase in lightning at higher latitudes. Therefore, the evaluation of future lightning is an important part of the overall evaluation of lighting in CanESM.

[Figure]

**Fig. 2.1:** (left) cloud-top height-based lightning for historical time period from Clark et al (2017). (right) Future changes to lightning using a cloud-top-height-based lightning scheme from Finney et al (2018).

[Figure]

**Fig. 2.2:** (left) Observation-based trend in lightning over the historical period from Qie et al (2020). (right) Future changes to lightning using an ice-flux lightning scheme from Finney et al (2018).

Therefore, we consider it highly relevant to include future projections of lightning and burned area since we want to be able to use CanESM to answer the question: How will wildland fires change in a warming climate?

This objective of our manuscript was encompassed in the second part of our title: "A new lightning scheme in Canada's Atmospheric Model, CanAM5.1: **Implementation, evaluation, and projections of lightning and fire in future climates**"

---

## Author Response (AR2)

Note from Editor:

Dear authors,
it comes to my attention that the "Code and Data Availability" section in your manuscript points out Git repository, which is not valid for scientific publication. Please read carefully the policy: https://www.geoscientific-model-development.net/policies/code_and_data_policy.html and make changes in this section accordingly. Publish your code in one of the appropriate repositories, only after that, your manuscript can be published in GMD.
Editor

Author's response:

Thank you. We have updated the Code and Data Availability to indicate a new location for the CanAM5.1 model code, which has been uploaded to Zenodo.